# Bridging the Training-Inference Gap in LLMs by Leveraging Self-Generated Tokens

**Zhepeng Cen**                                                    *zcen@andrew.cmu.edu*
*Carnegie Mellon University*

**Yao Liu**                                                        *yaoliuai@amazon.com*
*Amazon Web Services*

**Siliang Zeng**                                                   *zeng0176@umn.edu*
*University of Minnesota Twin Cities*

**Pratik Chaudhari**                                              *prtic@amazon.com*
*Amazon Web Services*

**Huzefa Rangwala**                                               *rhuzefa@amazon.com*
*Amazon Web Services*

**George Karypis**                                                *gkarypis@amazon.com*
*Amazon Web Services*

**Rasool Fakoor**                                                 *fakoor@amazon.com*
*Amazon Web Services*

**Reviewed on OpenReview:** *https://openreview.net/forum?id=pWSrm3oP8b*

## Abstract

Language models are often trained to maximize the likelihood of the next token given past tokens in the training dataset. However, during inference time, they are utilized differently, generating text sequentially and auto-regressively by using previously *generated* tokens as input to predict the next one. Marginal differences in predictions at each step can cascade over successive steps, resulting in different distributions from what the models were trained for and potentially leading to unpredictable behavior. This paper proposes two simple approaches based on model own generation to address this discrepancy between the training and inference time. Our first approach is Batch-Scheduled Sampling, where, during training, we stochastically choose between the ground-truth token from the dataset and the model's own generated token as input to predict the next token. This is done in an offline manner, modifying the context window by interleaving ground-truth tokens with those generated by the model. Our second approach is Reference-Answer-based Correction, where we explicitly incorporate a self-correction capability into the model during training. This enables the model to effectively self-correct the gaps between the generated sequences and the ground truth data without relying on an external oracle model. By incorporating our proposed strategies during training, we have observed an overall improvement in performance compared to baseline methods, as demonstrated by our extensive experiments using summarization, general question-answering, and math question-answering tasks.

## 1 Introduction

The common approach to training auto-regressive models is known as teacher forcing (Williams & Zipser, 1989). In this method, the ground truth token from the previous time step is utilized as the input for the

model at the current time step. This technique allows the model to learn relationships between tokens more effectively, facilitating faster convergence during the training process. While this training technique has been widely adopted in previous works (Cho et al., 2014; Gregor et al., 2014; Bahdanau et al., 2015; Vinyals et al., 2015; Parmar et al., 2018; Fakoor et al., 2017; 2018; 2020; Esser et al., 2020; Chang et al., 2022; Li et al., 2024; Liu et al., 2024), it can also lead to overfitting and undesirable behavior (Bengio et al., 2015; Bachmann & Nagarajan, 2024). In particular, when a model is solely trained on the provided ground-truth tokens, it may fail to behave reliably when encountering its own generations later, which can include unseen tokens. This occurs because, during inference, the model must rely solely on its own previous generations/predictions rather than the actual ground truth; hence, any small error can propagate through subsequent time steps, resulting in compounding errors and, therefore, unpredictable behavior. This issue is commonly known as exposure bias (Bengio et al., 2015; Ranzato et al., 2015; Schmidt, 2019; He et al., 2019).

Current transformer-based (Vaswani et al., 2017) methods for aligning large language models (LLMs) with human preferences, such as Reinforcement Learning from Human Feedback (RLHF), are also auto-regressive models (Ziegler et al., 2020; Ouyang et al., 2022; OpenAI, 2023; Gemini-Team, 2024). Supervised Fine-Tuning (SFT) is employed to fine-tune the pre-trained model using human demonstrations (utilizing teacher forcing) as the initial step of this alignment method. This fine-tuned model is then further refined using reinforcement learning through a reward model that serves as a proxy for human preferences (Ouyang et al., 2022). Since SFT is used as a standalone alignment method and also serves as the initialization for other steps in RLHF, such as the RL step that exclusively uses its own generations, having an SFT model that can utilize its own generations during training to develop tolerance for the shift between ground-truth data and its own generations seems both necessary and increasingly important. Therefore, it is important to develop an SFT model that is not solely trained on ground-truth data, which is more prone to overfitting and, consequently, to training-inference discrepancies, but also incorporates its own self-generated data during training to closely align with what it encounters during inference.

To bridge the discrepancies between how the model is trained and how it is used during inference, we propose two approaches that both leverage the model's own generations to address this issue. First, we train the model in a manner akin to how samples are generated during inference. Specifically, instead of relying solely on ground-truth tokens during training (teacher forcing), we expose the model to its own generations, adopting the scheduled sampling (Bengio et al., 2015) for LLMs but in an offline and batch manner, particularly during SFT training. While scheduled sampling and its variants have been popular with smaller models, especially recurrent-based ones (Lamb et al., 2016; Goyal et al., 2017; Li et al., 2020), they have not been widely adopted for LLMs due to their complexity and the practical challenges of incorporating them during training. We propose an offline and batch version of scheduled sampling called **Ba**tch-**sch**eduled Sampling (**BASH**), which is more practical and can be easily adopted during SFT training. Despite its effectiveness, one of the main problems with BASH is that the model's own generated tokens at each time step can deviate from the ground-truth tokens. When interleaved with ground-truth tokens, the resulting sequence can differ significantly from the ground truth, complicating training and ultimately leading to slower training time, which negatively impacts results. To address this, we introduce **R**eference-**A**nswer-based **C**orrection (**RAC**), where we explicitly incorporate a self-correction capability into the model. This approach resembles Dataset Aggregation (Ross et al., 2011) in imitation learning but employs a self-supervised objective without relying on an external oracle model.

To evaluate the effectiveness of our proposed approaches, we provide a comprehensive empirical comparison and ablation study of our method across a range of standard benchmark tasks, such as summarization (Stiennon et al., 2020) and general (Ding et al., 2023) and math question-answering tasks (Cobbe et al., 2021; Hendrycks et al., 2021). This evaluation is conducted in settings where we have access only to the human demonstrations data, which is primarily applicable to the SFT stage. Our results, based on win rates against the reference for the summarization task and length-controlled win rates (Dubois et al., 2024a) on the AlpacaEval 2.0 benchmark (Dubois et al., 2024b) for QA tasks, clearly demonstrate that our proposed approaches are effective in improving performance. Additionally, we demonstrate that initializing a model trained using our approach, followed by fine-tuning with preference data through a direct preference alignment method (Rafailov et al., 2024), leads to better results compared to initializing with a standard SFT model.

## 2 Background

Given a pre-trained auto-regressive language model parameterized by $\omega$, our objective is to fine-tune this model using human demonstration (a.k.a. expert) data[1] to ensure that it generates text aligned with the demonstration data. Consider a dataset $\mathcal{D} = \{(x_i, y_i)\}_{i=1}^N$ where $x_i$ and $y_i$ represent a query/prompt and its corresponding continuation, respectively. Each example is a sequence of tokens $x_i = (x_i^1, \ldots, x_i^T)$ and $y_i = (y_i^1, \ldots, x_i^L)$, where $T$ and $L$ indicate the lengths of the prompt and continuation, respectively[2].

To fine-tune this auto-regressive model with $\mathcal{D}$, we employ a maximum-likelihood objective, defined as follows:

$$\mathcal{J}_{\text{SFT}}(\theta) = \frac{1}{|\mathcal{D}|} \sum_{(x,y) \in \mathcal{D}} \sum_{j=0}^{L} \log p_\theta(y^j \mid x, y^{<j}) \tag{1}$$

where $\theta$ is the model parameters initialized from $\omega$, $y^{<j} = (y^0, y^1, \ldots y^{j-1})$, $y^0$ shows the beginning-of-sentence token, $L$ denotes length of the continuation sequence $y$, and $x = (x^1, \ldots, x^T)$ indicates a prompt of length $T$. To maintain consistency with current literature, we refer to this method as SFT (Ouyang et al., 2022).

### 2.1 Auto-regressive Generation

After the model is trained, we use it to perform conditional auto-regressive generation by specifying a prefix sequence $x^{1:T}$ (i.e. a prompt/query) and sampling the remaining sequence $z$ (i.e. continuation) one token at the time, using the prefix and the previously generated tokens as a context:

$$z^k \sim p_\theta(\cdot | x, z^{<k}), \quad k = 0, \cdots, K \tag{2}$$

where the current context looks as follows:

$$z_{\text{context}}^{T+k+1} = (\underbrace{x^1, \ldots, x^T}_{\text{prompt}}, \quad \underbrace{z^0, z^1, \ldots, z^k}_{\text{generated tokens so far}})$$

This process is repeated until the stopping condition is met (i.e., the maximum sequence length is reached or the end-of-sentence token is generated). Throughout the remainder of this paper, whenever generation is mentioned, we will utilize the auto-regressive approach explained above.

## 3 Methods

The objective function in Eq. (1) is known as teacher-forcing (Williams & Zipser, 1989) learning method for auto-regressive models, where it utilizes the *ground-truth* token from the previous time step as input to the model at the current time step. This can help the model learn more quickly, but it can also lead to overfitting. In particular, models trained exclusively on ground-truth data might exhibit inconsistent and undesirable behavior when faced with their own generated tokens during inference, especially if the generated token was not seen during training. Motivated by these challenges and to bridge the gap between training and inference, we propose two approaches in the following sections to address the shortcomings of the teacher-forcing method, making training resemble inference as closely as possible.

### 3.1 Batch-scheduled Sampling

**Scheduled sampling.** To align the model's behavior during training with how it functions during inference, and to account for its auto-regressive nature, we update Eq. (1) so that it consumes its own generated tokens in addition to the ground truth tokens during training, but in a controlled manner. To achieve this, the

---

[1]Note that demonstration data does not always need to come from humans, as it can also be synthetically generated from another model.

[2]To simplify notation, we drop the subscript $i$ from equations whenever it clears from the context.

---

**Algorithm 1** Batch-scheduled Sampling (BASH)

---

**Input**: Pre-trained model $\omega$, training datatset $\mathcal{D}$.

1: Initialize $\theta \leftarrow \omega$
2: **for** $k = 1, 2, \ldots, K_1$ **do**
3:     Sample mini-batch $\mathcal{B} = \{(x, y)\} \sim \mathcal{D}$
4:     $\nabla_\theta \mathcal{J}_{\mathrm{SFT}}(\theta) \leftarrow \nabla_\theta \frac{1}{|\mathcal{B}|} \sum_{(x,y) \in \mathcal{B}} \sum_j \log p_\theta(y^j \mid x, y^{<j})$
5:     $\theta \leftarrow \theta - \alpha \nabla_\theta \mathcal{J}_{\mathrm{SFT}}(\theta)$
6: **end for**
7: **for** iteration $1, 2, \ldots, H$ **do**
8:     Construct dataset $\mathcal{D}_s$ in offline manner as explained in Sec. 3.1
9:     **for** $k = 1, 2, \ldots, K_2$ **do**
10:         Sample mini-batch $\mathcal{B} = \{(x, y, \hat{y})\} \sim \mathcal{D}_s$
11:         $\nabla_\theta \mathcal{J}_{\mathrm{SFT}}(\theta) \leftarrow \nabla_\theta \frac{1}{|\mathcal{B}|} \sum_{(x,y) \in \mathcal{B}} \sum_j \log p_\theta(y^j \mid x, y^{<j})$
12:         $\nabla_\theta \mathcal{J}_{\mathrm{BASH}}(\theta) \leftarrow \nabla_\theta \frac{1}{|\mathcal{B}|} \sum_{(x,y,\hat{y}) \in \mathcal{B}} \sum_j \log p_\theta(y^j \mid x, \hat{y}^{<j})$
13:         $\theta \leftarrow \theta - \alpha \Big( \nabla_\theta \mathcal{J}_{\mathrm{SFT}}(\theta) + \nabla_\theta \mathcal{J}_{\mathrm{BASH}}(\theta) \Big)$
14:     **end for**
15: **end for**

**Output**: $\theta$.

---

scheduled sampling (SCS) method from Bengio et al. (2015) can be utilized. The goal of the scheduled sampling method is to stochastically include the model's generated tokens in an online manner during training:

$$\mathcal{J}_{\mathrm{SCS}}(\theta) = \frac{1}{|\mathcal{D}|} \sum_{(x,y) \in \mathcal{D}} \sum_{j=0}^{L} \log p_\theta(y^j \mid x, g^{<j}) \tag{3}$$

where $g^{<j}$ is a mixture of ground truth tokens and the model's own generations. To construct $g$, we first sample $z^j$ from the model output given the previous context $g^{<j}$ input: $z^j \sim p_\theta(\cdot|x, g^{<j})$. Then, $g^j$ is created by randomly selecting either $z^j$ or $y^j$ as the token with probability $\beta$:

$$g^j = \begin{cases} z^j & \text{with prob. } \beta \\ y^j & \text{otherwise} \end{cases} \tag{4}$$

Here, $\beta$ represents the mixing factor: when it equals 0, $g$ is exactly the same as $y$. However, when $\beta$ equals 1, $g$ becomes completely different from $y$, as every token is auto-regressively generated by the model, and the ground truth continuation tokens are completely discarded. Notice that, the context $g^{<j}$ in $\mathcal{J}_{\mathrm{SCS}}(\theta)$ is a function of the current parameter $\theta$, though we do not propagate the gradient through it in the standard SCS.

**Remark 1 (Scheduled sampling is not scalable).** While SCS (and its variants) has proven effective with small models (Bengio et al., 2015; Lamb et al., 2016; Goyal et al., 2017; Li et al., 2020), particularly recurrent-based networks, its computational complexity outweighs its benefits for LLMs. This is why it has not been widely adopted for training large models. Specifically, because LLMs require distributed training across many GPUs, switching between training and inference modes per tokens would not only significantly slow down the training but also lead to significant GPU under-utilization and memory related issues.

Mihaylova & Martins (2019) also highlighted the difficulty of applying scheduled sampling to transformer-based models and proposed structural changes to the transformer through a two-pass decoding strategy. In the first pass, model predictions are generated without accumulating gradients, and in the second pass, the generated data is mixed with the ground truth to update the model. Although this approach shows promise in some tasks, it has not been applied to LLMs. This is due to the required structural changes to the model's architecture and the discussed scalability challenges.

*Question:*
**Nelly is very pleased with the painting she bought at the auction. She tells her daughter that she outbid her rival Joe by paying $2000 more than thrice his bid. If Joe's bid was $160,000, at how much did Nelly get the painting?**

*SFT-model-generated Answer:*
**Nelly paid $2000 more than thrice Joe's bid, which means she paid 3 * $160000 + $2000 = 484000**
**The answer is: 484000**

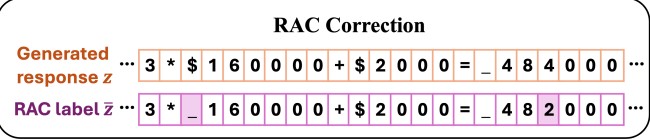

Figure 1: **How does RAC correct mistakes in model-generated responses?** In this example, SFT model makes a mistake in calculating $3 * \$160,000 + \$2,000$, as shown in yellow. However, RAC corrects the error by replacing the wrong token, 4, with the correct token, 2. This is achieved by forcing the model to fit $\bar{z}$ that differ from the original generated response (highlighted in purple), enabling it to self-correct. This example is based on the GSM8K dataset (Cobbe et al., 2021).

**Batch-scheduled Sampling.** To mitigate the limitations discussed in SCS for large models, we propose a simple yet effective offline approach. A new dataset, $\mathcal{D}_s = \{(x_i, y_i, \hat{y}_i)\}_{i=1}^M$, is created *offline* between training iterations by current model[3]. While $x_i$ remains identical to the original dataset $\mathcal{D}$, $\hat{y}_i$ represents a "mixed" continuation constructed by stochastically combining ground truth tokens with generated ones from the model, as described in the previous section (see Eq. (4)). It is important to emphasize that the offline and batch nature of BASH minimizes the cost of switching between training and inference times by creating $\mathcal{D}_s$ *only* between each iteration of training. In contrast, SCS switches between training and inference modes at every gradient step, resulting in significant slowdowns in training time and practical challenges related to distributed training[4]:

$$\mathcal{J}_{\text{BASH}}(\theta) = \frac{1}{|\mathcal{D}_s|} \sum_{(x,y,\hat{y}) \in \mathcal{D}_s} \sum_{j=0}^{L} \log p_\theta(y^j \mid x, \hat{y}^{<j}) \tag{5}$$

In practice, to balance optimizing the objective function using generated data and ground-truth data, we combine Eq. (1) and Eq. (5) (i.e. $\mathcal{J} = \mathcal{J}_{\text{sft}}(\theta) + \mathcal{J}_{\text{BASH}}(\theta)$), where the only difference is training data. Moreover, to minimize discrepancies between the current model and the one used for generation, this procedure can be repeated across different training iterations using the updated model from the most recent iteration. It is important to note that since the model can generate random tokens at the beginning of training, which complicates the training process, we start by first optimizing Eq. (1) alone. After a few iterations, we then begin including BASH. These steps are detailed in Algorithm 1.

It is worth noting that our method offers a distinct advantage over existing approaches, as it does not necessitate any modifications to the model structure, unlike methods such as Mihaylova & Martins (2019). Moreover, our method is scalable and can be applied to LLMs as shown in our experiments.

**Remark 2 (Parameter $\beta$ should be chosen to be small).** Depending on the value of $\beta$, there may be a distribution mismatch between the ground truth sequences $y$ and the mixed ones $g$. This gap can become larger as the value of $\beta$ approaches 1, as $g$ increasingly differs from $y$, not just in terms of a few tokens, but at the sequence level. Also, since scheduled sampling can result in a biased estimator (Huszár, 2015; Lamb et al., 2016), the training can get more harder, as the model needs to learn to fit the ground truth data with the altered continuations which are far from each others. Therefore, it is important to keep the value of $\beta$ small to avoid making the optimization problem harder to solve [5].

---

[3]If we drop $\hat{y}_i$ from $\mathcal{D}_s = \{(x_i, y_i, \hat{y}_i)\}$, we retrieve the original $\mathcal{D}$. Practically, this is useful during training, as $\mathcal{D}_s$ can also function as the original $\mathcal{D}$ (when $\hat{y}_i$ is dropped), eliminating the need for separate data loaders to train both SFT and BASH.

[4]See https://huggingface.co/docs/transformers/en/performance to learn about the challenges of distributed training.

[5]In our experiments in this paper, we select $\beta$ to be equal to 0.2.

### 3.2 Reference-Answer-based Correction

One of the main problems with scheduled sampling, which mixes ground truth tokens $y^j$ and the model's own generations $z^j$, is that the resulting sequence $g$ can diverge significantly from the ground truth sequence $y$. This means that the generated sequence $g$ not only differs in individual tokens but also conveys a context and meaning that deviate substantially from $y$. Importantly, since the scheduled sampling approach (both online and offline) results in a biased estimator (see Remark 2), such significant discrepancies between the ground truth and generated sequences can further complicate training and ultimately lead to slower training/convergence time. To give the model ability to recover in such scenarios, we propose **R**eference-**A**nswer-based **C**orrection (RAC), where we explicitly incorporate a self-correction capability into the model.

To build RAC, we first construct a dataset $\mathcal{D}_r = \{(x_i, y_i, z_i, \bar{z}_i)\}_{i=1}^N$ in an offline manner. Here, $z$ denotes a sequence of model's own generated tokens (see Eq. (2)) and $\bar{z}$ denotes a new target/label sequence, also composed of the model's own generated tokens, which is constructed by greedy sampling from model $\theta$ at each time step:

$$\bar{z}^j = \arg\max_z p_\theta(z | f(x, y), z^{<j})$$

where $f$ is a prompt template[6]. The reason for constructing $\bar{z}^j$ in this way is to enable the model to leverage its self-correction capability by incorporating the ground-truth answer $y$ into the input context. Hence, the maximum-likelihood objective for RAC can be written as follows:

$$\mathcal{J}_{\mathrm{RAC}_0}(\theta) = \frac{1}{|\mathcal{D}_r|} \sum_{(x,y,z,\bar{z})\in\mathcal{D}_r} \sum_{j=0}^L \log p_\theta(\bar{z}^j \mid x, z^{<j}), \tag{6}$$

In RAC, the model attempts to maximize the likelihood of correction label $\bar{z}^j$ conditioned on the original prompt $x$ and its own generated tokens $z$. This contrasts with BASH, where $\hat{y}$ is used instead of $z$, and the target token is always $y^j$, i.e., $p_\theta(y^j \mid x, \hat{y}^{<j})$. It is important to note that $\bar{z}^j$ is generated using greedy sampling, conditioned on $f(x, y)$ and $z$. Conditioning on $f(x, y)$ rather than just $x$ allows for additional context and guidance during the generation process.

One issue that arises in Eq. (6) is when $\bar{z}^j$ becomes identical to the generated token $z^j$. When this occurs, it can lead to model collapse (Shumailov et al., 2024), as the model is likely to only learn trivial solutions. To mitigate this issue, we mask out such tokens and rewrite the objective function as follows:

$$\mathcal{J}_{\mathrm{RAC}}(\theta) = \frac{1}{|\mathcal{D}_r|} \sum_{(x,z,\bar{z})\in\mathcal{D}_r} \sum_{j=0}^L \mathbf{1}(\bar{z}^j \neq z^j) \log p_\theta(\bar{z}^j \mid x, z^{<j}), \tag{7}$$

Similar to BASH's algorithm, we combine Eq. (1) and Eq. (7) by first optimizing Eq. (1) alone for several iterations to avoid relying on model generation at the start of training. After this initial phase, we then include RAC objective function and jointly optimize with SFT, i.e. $\mathcal{J} = \mathcal{J}_{\mathrm{sft}}(\theta) + \mathcal{J}_{\mathrm{RAC}}(\theta)$. These steps are detailed in Algorithm 2. Fig. 1 shows an example of how the self-correction capability of RAC helps in producing correct results.

## 4 Experiments

In this section, we present a comprehensive empirical comparison of our proposed methods across a range of standard benchmark tasks, including summarization and question answering (QA). These results show that the effectiveness and robustness of our approaches across different settings. See also Appendix A.2 for a description of our experiments and Appendix A.3 for more ablation studies.

---

[6]One example for prompt template $f(x, y)$ is "I will give you a question and a reference response. You need to give a new response based on the reference response. Question: $x$. Reference response: $y$". See Appendix A.1 for the actual templates used in experiments

---

**Algorithm 2** Reference-Answer-based Correction (RAC)

---

**Input**: Pre-trained model $\omega$, training dataset $\mathcal{D}$.

1: Initialize $\theta \leftarrow \omega$
2: **for** $k = 1, 2, \ldots, K_1$ **do**
3:      Sample mini-batch $\mathcal{B} = \{(x, y)\} \sim \mathcal{D}$
4:      $\nabla_\theta \mathcal{J}_{\text{SFT}}(\theta) \leftarrow \nabla_\theta \frac{1}{|\mathcal{B}|} \sum_{(x,y) \in \mathcal{B}} \sum_j \log p_\theta(y^j \mid x, y^{<j})$
5:      $\theta \leftarrow \theta - \alpha \nabla_\theta \mathcal{J}_{\text{SFT}}(\theta)$
6: **end for**
7: **for** iteration $1, 2, \ldots, H$ **do**
8:      $\mathcal{D}_r \leftarrow \varnothing$
9:      **for** $i = 1, 2, \cdots, N$ **do**
10:        Generate model's response $z_i$ by $z_i^j \sim p_\theta(\cdot | x_i, z_i^{<j}), j = 1, 2, \ldots$
11:        Generate RAC label $\bar{z}_i$ by $\bar{z}_i^j = \arg\max_z p_\theta(z | f(x_i, y_i), z_i^{<j}), j = 1, 2, \ldots$
12:        $\mathcal{D}_r \leftarrow \mathcal{D}_r \cup \{(x_i, y_i, z_i, \bar{z}_i)\}$
13:      **end for**
14:      **for** $k = 1, 2, \ldots, K_2$ **do**
15:        Sample mini-batch $\mathcal{B} = \{(x, y, z, \bar{z})\} \sim \mathcal{D}_r$
16:        $\nabla_\theta \mathcal{J}_{\text{SFT}}(\theta) \leftarrow \nabla_\theta \frac{1}{|\mathcal{B}|} \sum_{(x,y) \in \mathcal{B}} \sum_j \log p_\theta(y^j \mid x, y^{<j})$
17:        $\nabla_\theta \mathcal{J}_{\text{RAC}}(\theta) \leftarrow \nabla_\theta \frac{1}{|\mathcal{B}|} \sum_{(x,z,\bar{z}) \in \mathcal{B}} \sum_j \mathbf{1}(\bar{z}^j \neq z^j) \log p_\theta(\bar{z}^j \mid x, z^{<j})$
18:        $\theta \leftarrow \theta - \alpha\Big(\nabla_\theta \mathcal{J}_{\text{SFT}}(\theta) + \nabla_\theta \mathcal{J}_{\text{RAC}}(\theta)\Big)$
19:      **end for**
20: **end for**

**Output**: $\theta$.

---

## 4.1 Setups

### 4.1.1 Benchmark Tasks and Evaluation Metrics

**Summarization task**. We use OpenAI TL;DR dataset (Stiennon et al., 2020) for this task, which includes posts from Reddit forum and their corresponding summaries from human labelers. We evaluate performance by calculating the win rate against the reference summary and reporting Rouge F1 scores (Lin, 2004) on its test set.

**General QA task**. For this task, we use the Ultrachat-200K dataset, a high-quality 200K subset of the Ultrachat corpus (Ding et al., 2023), which contains approximately 1.4 million general QA dialogues generated by ChatGPT (3.5) Turbo API. We evaluate performance using the length-controlled (LC) win rate (Dubois et al., 2024a) on the AlpacaEval 2.0 benchmark (Dubois et al., 2024b).

**Math QA task**. We also compare the language model's ability in mathematical calculation and reasoning. To do this, we use two commonly used math QA datasets: GSM8K (Cobbe et al., 2021) and MATH (Hendrycks et al., 2021), and evaluate the accuracy on their respective test sets.

### 4.1.2 Baselines Methods

Considering the setting of this paper, which applies in cases where access is limited *only* to human demonstration data, we compare our methods against existing approaches that also rely exclusively on demonstration data: SFT (Ouyang et al., 2022), NEFTune (Jain et al., 2023), and SPIN (Chen et al., 2024). These methods represent a strong set of supervised approaches based on demonstration data. NEFTune introduces noise to input token embeddings to improve the model's robustness, and SPIN utilizes self-replay to ensure that the generated outputs remain indistinguishable from the reference demonstrations.

### 4.1.3 Training

We use Pythia-1B as the pre-trained model for the summarization experiments and Mistral-7B-v0.1 for the general QA and math QA experiments. Additionally, we use Mistral-7B-sft-beta as the SFT model for the general QA task, as it is a fine-tuned version of Mistral-7B-v0.1[7]. Therefore, we report its performance as SFT model and use it to fine-tune SPIN and our methods. For the summarization and math QA tasks, following SPIN (Chen et al., 2024), we first perform SFT on the pre-trained models and then train SPIN, BASH, and RAC on top of them. See Appendix A.2 for more details on the baselines and our methods.

## 4.2 Main results

### 4.2.1 Summarization Task

For this task, we train SFT and NEFTune for two epochs, starting from the Pythia-1B base model, and then continue fine-tuning SFT model with SPIN, BASH, and RAC for one more epoch, as no noticeable improvements were observed beyond that point. We compare their performance based on the win rate of the generated outputs against the reference responses on the test set, evaluated by the GPT-4 Turbo model. The evaluation prompt template is provided in Appendix A.1. To ensure more comprehensive results, we repeat the training with three different random seeds and report the win rates along with the Rouge F1 scores (Lin, 2004) in Table 1.

Table 1: **Comparison of the performance (higher is better) of our methods against others on the summarization task**. We train each method using three different seeds and report the average win rate across them in addition to Rouge scores. The gray denotes the standard deviations.

|              | win rate (%)       | Rouge 1   | Rouge 2   | Rouge L   |
| ------------ | ------------------ | --------- | --------- | --------- |
| SFT          | $27.03_{\pm 0.38}$ | 31.69     | 11.02     | 24.53     |
| NEFTune      | $27.21_{\pm 0.40}$ | **31.97** | 11.16     | 24.73     |
| SPIN         | $21.07_{\pm 1.31}$ | 30.46     | 10.09     | 22.55     |
| BASH (ours)  | $\mathbf{28.12}_{\pm 0.43}$ | **32.00** | **11.27** | **24.89** |
| RAC (ours)   | $28.02_{\pm 0.39}$ | **32.01** | 11.20     | 24.80     |

As Table 1 shows, although our methods outperform the others, the improvement is not particularly significant. This could be attributed to the nature of the summarization task, where the prompts and queries are quite long, but the generated responses are very brief. Since our method operates directly on the response space, the relatively short length of the responses means that there is less to correct during the generation steps. Despite the inherent limitations of the task, our method still improves the base models, demonstrating its applicability across different tasks.

### 4.2.2 General QA task

Following SPIN (Ouyang et al., 2022), we randomly sample 50K prompts from the full training set to generate offline datasets $\mathcal{D}_s$ and $\mathcal{D}_r$ for our methods and then use these datasets to train our BASH and RAC as explained in Sec. 3.

To provide a comprehensive view of the results, we use the length-controlled win rate on AlpacaEval 2.0 as our evaluation metric, which compares the generated outputs against those from GPT-4, following the standard AlpacaEval prompt template. For each method, we evaluate performance using two types of generation: 1) Sampling-based generation with a temperature of 0.7, aligning with the default settings of the Mistral-7B model and its derivative, Zephyr-7B-Beta (Tunstall et al., 2023), on AlpacaEval[8]. Given the inherent randomness in single generation evaluations, we repeat the generation three times and report the average evaluation for each generation. 2) Greedy generation, which selects the next token by choosing the one with the highest probability.

---

[7] See https://huggingface.co/HuggingFaceH4/mistral-7b-sft-beta

[8] See https://github.com/tatsu-lab/alpaca_eval/blob/main/src/alpaca_eval/models_configs/zephyr-7b-beta/configs.yaml.

Table 2: **Comparison of the performance (higher is better) of our methods against others in the General QA task using the length-controlled win rate on the AlpacaEval 2.0 benchmark.** These results shows that our method consistently stands out, maintaining its effectiveness across different generation strategies.

| | | Temperature=0.7 (%) | | | | Greedy (%) |
|---|---|---|---|---|---|---|
| | | Generation 1 | Generation 2 | Generation 3 | Average | |
| SFT | | 7.90 | 8.45 | 7.74 | $8.03_{\pm 0.30}$ | 7.06 |
| NEFTune | | 7.60 | 6.99 | 7.56 | $7.38_{\pm 0.28}$ | 6.64 |
| SPIN | iter-1 | 8.76 | 9.27 | 9.41 | $9.15_{\pm 0.28}$ | 8.66 |
| | iter-2 | 8.44 | 8.89 | 8.17 | $8.50_{\pm 0.30}$ | 7.64 |
| BASH (ours) | iter-1 | 8.77 | 8.56 | 8.48 | $8.60_{\pm 0.12}$ | 7.31 |
| | iter-2 | 8.73 | 9.38 | 9.10 | $9.07_{\pm 0.27}$ | 8.42 |
| RAC (ours) | iter-1 | 9.95 | 9.15 | 9.49 | $9.53_{\pm 0.33}$ | 9.04 |
| | iter-2 | **10.68** | **10.54** | **9.89** | $\mathbf{10.37}_{\pm 0.34}$ | **9.41** |

Table 3: **Comparison of the test accuracy (higher is better) of our methods against baseline algorithms on GSM8K and MATH tasks, averaged across three different generations used for evaluation**.

| | GSM8K | MATH |
|---|---|---|
| SFT | $56.76_{\pm 0.09}$ | $13.11_{\pm 0.12}$ |
| NEFTune | $55.85_{\pm 0.38}$ | $12.13_{\pm 0.03}$ |
| SPIN | $46.31_{\pm 0.97}$ | $6.85_{\pm 0.08}$ |
| BASH (ours) | $\mathbf{60.22}_{\pm 0.31}$ | $\mathbf{14.59}_{\pm 0.09}$ |
| RAC (ours) | $59.41_{\pm 0.73}$ | $13.75_{\pm 0.17}$ |

As shown in Table 2, our RAC achieves the highest win rate in both sampling-based and greedy generation settings. Additionally, BASH also demonstrates consistent improvements compared to other methods, except for SPIN. It is important to note that our methods show nearly monotonic improvements and consistent behavior across different iterations, unlike methods such as SPIN [9]. This consistency highlights the applicability of our approach beyond a single iteration, suggesting that it can be effectively utilized with larger models and in more iterations, resulting in consistent improvements over time.

### 4.2.3 Math QA task

In this setting, we train SFT and NEFTune for two epochs, starting from the Mistral-7B-v0.1 base model, and then continue fine-tuning SFT model with SPIN, BASH, and RAC for one more epoch. We prompt the model to generate answers in the GSM8K and MATH test sets with query template attached in Appendix A.1. We then calculate the strict match accuracy of the generated answers with the ground truth. We use sampling-based generation with a temperature of 0.1, repeating the generation three times, and report the average accuracy in Table 3. As the results show, BASH and RAC outperform other methods on both GSM8K and MATH benchmarks. These results further demonstrate the effectiveness of our methods across diverse set of tasks.

### 4.2.4 Alignment with preference data

In this experiment, we further demonstrate that our model can serve as a more effective initialization than standard SFT throughout the alignment pipeline. Specifically, we show that using our method to initialize a

---

[9]We observe a decrease in the LC win rate of SPIN in the second iteration, primarily due to a significant increase in generation length while the content remains similar. Since the length-controlled win rate accounts for output length, this results in a performance drop. Similarly, the win rate for greedy generation is lower than for random sampling at a temperature of 0.7, as greedy generation tends to be more verbose.

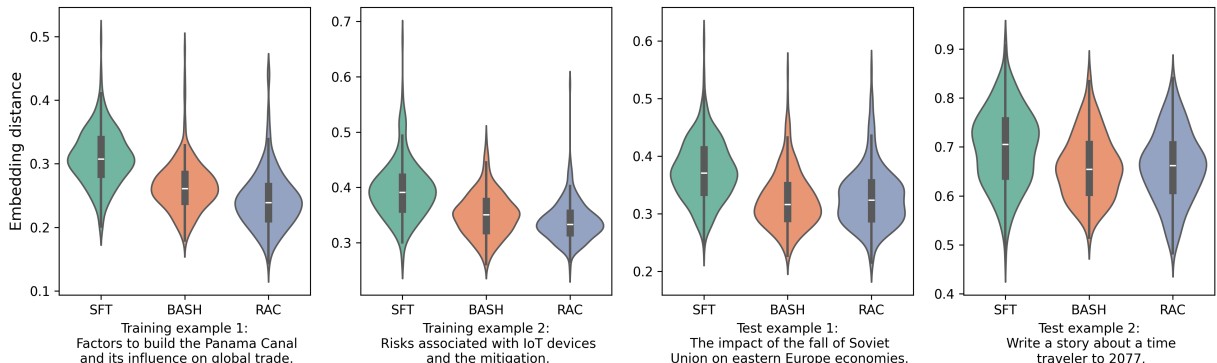

Figure 2: **Visualization of the embedding distance between generated and reference responses.** The left two figures are based on queries from the training set of the UltraChat-200K dataset, while the right two figures are from the test set. Corresponding queries for each figure are summarized at the bottom, with full queries available in Appendix A.3. We generate 256 responses from models and compute their embedding distances to the reference responses. Each violin plot includes an inner box plot that displays the maximum, third quartile, median (indicated by a white line), first quartile, and minimum distances, while the shape of the violin represents the estimated probability density of the embedding distance.

model, followed by fine-tuning with preference data via DPO (Rafailov et al., 2024), leads to better results than initializing with a standard SFT model. For this, we closely follow the settings of the zephyr-7b-beta model (Tunstall et al., 2023) and use the preprocessed UltraFeedback dataset (Cui et al., 2024) as the preference data. The results of this experiment, presented in Table 4, illustrate the importance of our approach in enabling downstream alignment methods to achieve improved performance when initialized with our method instead of standard methods like SFT.

Table 4: **Comparison of the performance (higher is better) of DPO initialized with our methods versus others on the AlpacaEval 2.0 benchmark.** These results clearly demonstrate the effectiveness of our method and provide further evidence of its applicability throughout the alignment pipeline.

|  | Temperature=0.7 (%) | | | | Greedy (%) |
| --- | --- | --- | --- | --- | --- |
|  | Generation 1 | Generation 2 | Generation 3 | Average | |
| SFT + DPO | 14.17 | 14.74 | 13.28 | $14.06_{\pm 0.60}$ | 14.22 |
| BASH + DPO | 14.66 | 13.34 | 13.61 | $13.87_{\pm 0.57}$ | 14.39 |
| RAC + DPO | **16.40** | **15.21** | **15.95** | $\mathbf{15.85}_{\pm 0.49}$ | **15.35** |

### 4.3 Qualitative Analysis

**Visualization**. To illustrate how our methods address the training-inference gap in autoregressive model training, we evaluate the discrepancy between model-generated responses and reference responses on the UltraChat dataset. We begin by selecting two queries from both the training and test sets and then use models fine-tuned with different methods to generate 256 responses by sampling with a temperature of 0.7. To measure the deviation of the generated responses from the reference responses, we first utilize Sentence Transformer[10] (Reimers & Gurevych, 2019) to extract the embeddings for each response. We then compute the distance between the embeddings of each generated response and its corresponding reference response, which we refer to as sentence distance. The distribution of these distances is visualized in Fig. 2. As shown in these plots, BASH and RAC reduce the discrepancy between the generated responses and the reference ones, except for the question with very open answers such as "write a story" in test example 2, indicating that our approach effectively narrows the gap between training and inference time.

---

[10]We use https://huggingface.co/sentence-transformers/all-mpnet-base-v2.

## 5    Conclusion

In this paper, we propose principled methods to bridge the gap between training and inference time in LLMs by leveraging self-generated tokens. Specifically, this gap arises from the training strategy where the model uses the ground truth token from the previous time step as the input for the model at the current time step. While this strategy is effective during training, the model must rely on its own predictions as input for subsequent steps during inference. This reliance on its own predictions can lead to error accumulation and degraded performance, particularly when the generated sequences deviate from the conditions encountered during training. Scheduled sampling has emerged as an alternative approach, where the model is gradually exposed to its own generated tokens during training, thus more closely simulating the conditions of inference. Despite its effectiveness, especially with smaller models and particularly in recurrent models, this approach has not been successfully adopted in training LLMs due to the computational and practical complexities of incorporating it into the training process. However, our proposed methods are specifically tailored for LLMs without requiring structural changes to the model. Specifically, our approaches, BASH and RAC, incorporate the model's own generations in an offline manner, allowing training that closely mirrors the inference process. Moreover, RAC also builds in self-correction capability during training, which becomes critical when the model's own generations deviate significantly from the ground truth. These methods can be easily integrated into the training of LLMs without requiring changes to the training process or model architectures, which was not the case with previous methods. Through a series of comprehensive experiments, we demonstrate that our method outperforms existing approaches in both summarization and question answering (QA) tasks. The results indicate that by aligning the training process with the conditions of inference, we can enhance the model's performance and reliability, ultimately leading to more accurate and contextually appropriate results.

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

# A  Experiment Details

## A.1  Prompt Templates

**RAC prompt template** $f(x, y)$ **in summarization task**. In experiment of summarization task, $x$ is the original post and $y$ is the reference summary from the TL;DR dataset. we will use the below template to generate RAC label.

> [POST]: x
> [REFERENCE SUMMARY]: y
> Re-write a new summary of the post and cover the main content in the reference summary.
> TL;DR:

**RAC prompt template** $f(x, y)$ **in general QA and math QA task**. In experiment of QA tasks, $x$ is query and $y$ is reference answer from the datasets. We will use the below template to generate RAC label. $\langle/s\rangle$ is the special token indicating the end of sentence.

> $\langle|system|\rangle$
> You are a helpful assistant to answer user's question. Your will be given both a question and reference response. You need to give a new response to the question and contain the main content in the reference response.$\langle/s\rangle$
> $\langle|user|\rangle$
> x
> Answer this question based on the following reference response: y$\langle/s\rangle$
> $\langle|assistant|\rangle$

**Evaluation prompt template in summarization task**. We use the below template to evaluate the win rate of generated summary against the reference summary with CoT. We also randomly shuffle the order between generated and reference summary to reduce the evaluation bias. In experiment, we will replace {post} by the post from test set and replace {summary A}, {summary B} by the shuffled generated and reference summaries.

> Which of the following summaries does a better job of summarizing the most important points in the given forum post, without including unimportant or irrelevant details? Judge based on accuracy, coverage, and coherence.
> [post]
> {post}
> [summary A]
> {summary A}
> [summary B]
> {summary B}
> Instructions:
> FIRST provide a one-sentence comparison of the two summaries, explaining which you prefer and why.
> SECOND, on a new line, state only A or B to indicate your choice. Your response should use the format:
> Comparison: one-sentence comparison and explanation
> Preferred: A or B

**Evaluation prompt template in math QA task**. We use the template below to test the accuracy of the generated answer. Unlike other benchmarks, we use zero-shot (i.e., no QA examples are provided before query) and CoT (Wei et al., 2022) to prompt the language model to generate answers. In the experiment, we will replace {query} by the questions from the test set.

> $\langle|system|\rangle$
> $\langle/s\rangle$
> $\langle|user|\rangle$

> {query}
> Let's think step by step.⟨/s⟩
> ⟨|assistant|⟩

## A.2   Hyperparameters and More Experiment Settings

**More implementation details**.

We implement baselines and our methods based on two codebases: summarize-from-feedback[11] (for summarization task) and Alignment-Handbook[12] (for general QA and math QA tasks), which use DeepSpeed ZeRO (Rajbhandari et al., 2020) for higher training efficiency and less computation overhead. In summarization task, we generate the response from the fine-tuned model with temperature=0.01 for win rate evaluation following the codebase used.

We train SFT models for summarization and math QA tasks from the corresponding base pretrained models. In particular, we do not pack data during SFT training, which is introduced in the T5 model (Raffel et al., 2020) and is a default option in alignment-handbook repository. For NEFTune, we adopt the noise scale $\alpha = 5$ in experiment. For SPIN, we use the official implementation[13] and follow all training settings to generate data and fine-tune based on the SFT model in general QA and math QA tasks. The only difference is that we adopt Mistral-7B-sft-beta as the base model. In summarization task, we implement SPIN ourselves to be compatible with the summarization code repository. We also search the hyperparameter $\beta$ of SPIN among $[0.1, 0.5, 1.0, 5.0]$ and report the highest win rate. Note that the enumeration starts from 0 in the SPIN paper, while ours starts from 1. Therefore, iteration 0 in the SPIN paper corresponds to iteration 1 in our paper.

**Hyperparameters**.

We attach the hyperparameters used in the experiment in table 5.

Table 5: **Hyper-parameters used for experiments.**

|  | Summarization | General QA | Math QA |
| --- | --- | --- | --- |
| base pretrained model | pythia-1B | Mistral-7B-v0.1 | Mistral-7B-v0.1 |
| precision | bfloat16 | bfloat16 | bfloat16 |
| optimizer | AdamW | AdamW | AdamW |
| learning rate | $3 \times 10^{-6}$ | $5 \times 10^{-6}$ | $5 \times 10^{-6}$ |
| learning rate warmup steps | no warmup | 10% | 10% |
| learning scheduler | cosine | cosine | cosine |
| global batch size | 512 | 512 | 512 |
| SFT training epoch | 2 | / | 1 |
| training iteration (BASH&RAC) | 1 | 2 | 1 |
| training epochs in each iteration (BASH&RAC) | 1 | [1, 2] | 1 |
| mixture coefficient $\beta$ in BASH generation | 0.2 | 0.2 | 0.2 |

## A.3   More Experiment Results

**Queries used in Fig. 2 for embedding distance computation**.

Query of training example 1:

> What factors influenced the decision to build the Panama Canal, and how did it transform global trade and transportation?

---

[11]We implement algorithms based on OpenAI summarize-from-feedback and its clean-up version.
[12]See link of alignment-handbook.
[13]See link of SPIN.

Query of training example 2:

> What are the risks associated with IoT devices, and how can they be mitigated?

Query of test example 1:

> How did the fall of the Soviet Union impact the economies of Eastern Europe?

Query of test example 2:

> Using vivid imagery and descriptive language, write a compelling short story about a time traveler who finds themselves transported to the year 2077. Explore the world of the future, including advanced technology, cultural and societal changes, and environmental shifts. Consider adding a twist or unexpected turn to the plot to keep your audience engaged. Your story should be between 500-1000 words in length and should captivate the reader from beginning to end.

**The performance comparison with different dataset sizes**.

In this experiment, we compare our methods with baselines on training datasets with different sizes in the summarization task. Specifically, we randomly choose a subset of training data with proportion $10\%, 20\%, 50\%, 100\%$. Then we keep all other settings the same and train the model for the same steps as the full dataset training. For example, when training on the 20% data, we will train for 10 epochs for SFT and 5 epochs for BASH or RAC (note that for the full data setting, we train 2 epochs for SFT and 1 epoch for BASH or RAC). We still report the win rate against the reference summary on the whole test set using the same evaluation prompt.

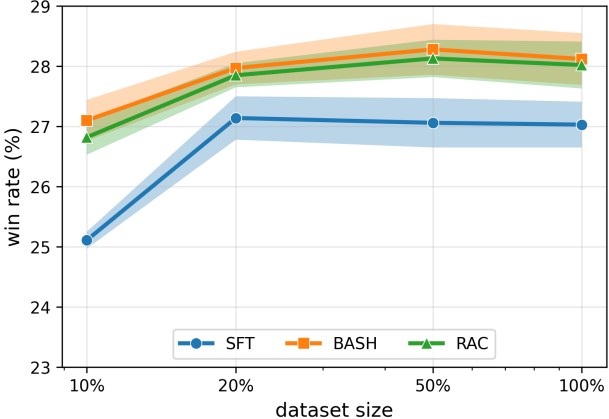

Figure 3: **The win rates of summarization task with different dataset sizes.** Each results are averaged over three seeds. In each seed, the subset of trainig data is different and we train model on the different subset for SFT, BASH and RAC. The win rate is evaluated on the whole test set.

The final results are reported in Fig. 3. We observe no notable performance drop when the dataset size exceeds 20%. When the dataset size becomes smaller, teacher-forcing-style SFT training suffers from the data insufficiency, leading to a more severe distribution shift between (Fakoor et al., 2024) training and inference. In this case, our methods can mitigate this issue and exhibit a relatively smaller win rate decrease compared to SFT.

**The performance of SFT training for more steps**.

We train the SFT fine-tuned model by SFT for more steps as an ablation. We also compare its LC win rate on AlpacaEval 2.0 with our methods in Fig. 4. More SFT training steps improve the performance of model on general QA task. However, there is a performance drop at the third epoch. On the contrary, our

methods ensure a monotonic increase in LC win rate and can consistently outperform the SFT baseline in each iteration.

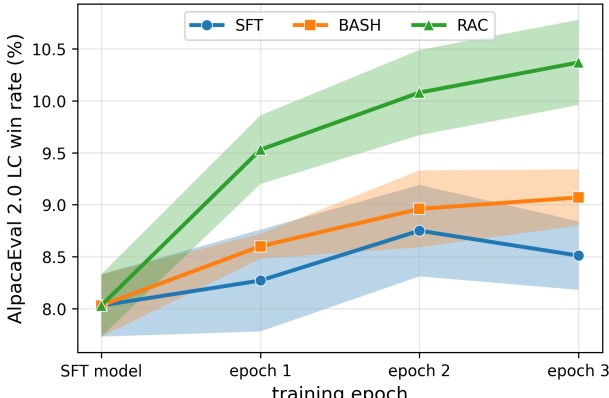

Figure 4: **The AlpacaEval 2.0 LC win rates comparison of our methods and SFT.** Each results are averaged over three generations. We leverage SFT on the UltraChat dataset to continue to train the existing SFT model and compare its performance with our methods. In the figure, the epoch 1 corresponds to the first iteration and epoch 2&3 correspond to second iteration for BASH and RAC. In the beginning of each iteration, we will offline generate BASH sequences or RAC labels by current model.

**The performance of training without combining the SFT loss**.

Table 6: **LC win rate comparison w.o. combining the SFT loss.** We evaluate the average of three sampled generation with temperature=0.7. The gray denotes the standard deviation of three evaluations.

|  | with SFT loss | w.o. SFT loss |
|---|---|---|
| SFT | $8.03_{\pm 0.30}$ | |
| BASH | $8.06_{\pm 0.46}$ | $9.07_{\pm 0.27}$ |
| RAC | $3.24_{\pm 0.48}$ | $10.37_{\pm 0.34}$ |

As shown in Table 6, there will be a large performance drop if we do not include the SFT loss in training. One potential reason is that the learning objective of BASH or RAC is a biased estimator (Lamb et al., 2016) of the expert language model, which can be viewed as the underlying model of SFT dataset $\mathcal{D}$. During training, the language model is enforced to fit the labels conditioned on a student input distribution significantly shifts from the one in the SFT objective in Eq. (1). Therefore, we include SFT loss, an unbiased behavioral cloning objective, to train the LM to imitate the underlying expert model of the dataset.

**The performance comparison between Scheduled sampling (SCS) and Batch-scheduled sampling (BASH)**.

Table 7: **Comparison of win rate on the summarization task**. We compare the performance

|  | win rate (%) | Rouge 1 | Rouge 2 | Rouge L |
|---|---|---|---|---|
| SFT | 27.03±0.38 | 31.69 | 11.02 | 24.53 |
| SCS | 28.24±0.46 | 32.03 | 11.22 | 24.95 |
| BASH | 28.12±0.43 | 32.00 | 11.27 | 24.89 |

Table 7 shows that scheduled sampling (SCS) achieves a win rate similar to BASHas expected. However, BASHis much more computationally efficient than SCS, which is crucial in the practical implementation of LLM training. To provide further perspective on the effectiveness of our approach, we conduct experiments comparing the computational overhead of SCS and BASH using 1B and 7B models.

Table 8: **Computation overhead comparison between SCS and BASHwith pythia-1B model on the summarization task**. We compare the performance on an 8xA6000 (48G) machine. For fair comparison, we set the local batch size as 16 and gradient accumulation step as 4 for all experiments. In this case, the global batch size is $16 * 4 * 8 = 512$. The computation time of SCS includes both (online) data generation and model training.

|  | SCS | BASH (offline data generation) | BASH (training) |
|---|---|---|---|
| Computation time for 1 global batch | ~20.8s | ~0.9s | ~4.4s |
| Memory usage of each GPU | ~35GB | ~17GB | ~28GB |

Table 9: **Computation overhead comparison between SCS and BASHwith Mistral-7B model on the general QA task**. We compare the performance on an 8xA6000 (48G) machine. For fair comparison, we set the local batch size as 8 and gradient accumulation step as 8 for all experiments. In this case, the global batch size is $8 * 8 * 8 = 512$. The computation time of SCS includes both (online) data generation and model training.

|  | SCS | BASH (offline data generation) | BASH (training) |
|---|---|---|---|
| Computation time for 1 global batch | >3600s | ~370s | ~75s |
| Memory usage of each GPU | ~46GB | ~22GB | ~36GB |

As these results show, BASH requires significantly less GPU memory and computation time than SCS, making it more practical in LLM training.

## A.4    More analysis of RAC

In this section, we first demonstrate how the RAC correction mechanism operates, followed by an example illustrating its failure case.

In Fig. 5, we present an example from GSM8K training set to illustrate how RAC leverages in-context learning to corrects the error in model-generated response by appending the reference response in the query.

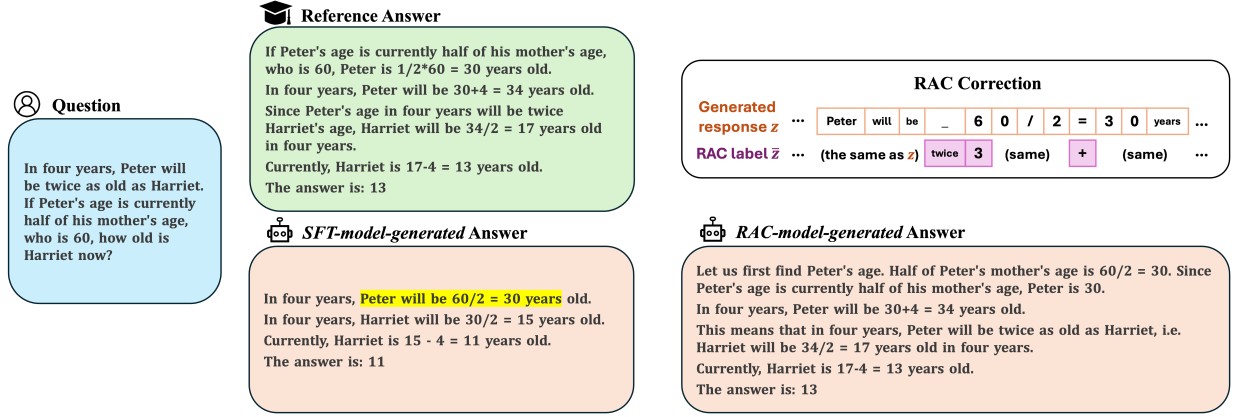

Figure 5: **How does RAC correct mistakes in model-generated responses?** In this example, SFT model incorrectly calculates Peter's age in the initial step, deriving it as $60/2 = 30$. However, the model fine-tuned with RAC produces the correct result. Specifically, the SFT model's incorrect calculation leads to an age of 30, while the ground truth is 34. To address this error, RAC labels the next token after "Peter will be 60/2" as "+" instead of "=", guiding the model towards the correct computation. After training with RAC, the model successfully calculates Peter's age accurately, resulting in the correct answer.

We also present a failure case of RAC correction. Despite being augmented by the reference answer, RAC is primarily trained on a self-generated reasoning trajectory and may struggle to identify complex reasoning errors in the model-generated response. In Fig. 6, the RAC model attempts to correct the final answer

directly; however, the token-wise correction is inconsistent with the context of the preceding tokens and fails to address the underlying errors in the generated response.

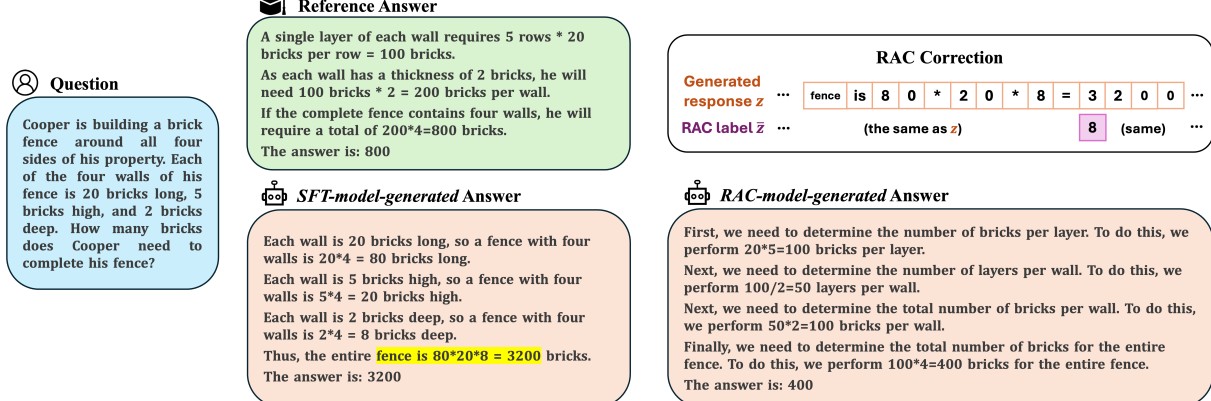

Figure 6: **An example of RAC failure**. Given the reference answer, RAC model labels the next token of $80 * 20 * 8 =$ as 8 to match the ground-truth answer 800. However, such correction is not consistent with the previous context. Meanwhile, RAC correction fails to find the deeper reasoning error that the entire fence is not 80 long, 20 high or 8 deep and does not correct it when the generated answer multiplies them together.

