# OpenReview forum: "Bridging the Training-Inference Gap in LLMs by Leveraging Self-Generated Tokens"
_TMLR — Accepted by TMLR_

### Review · Reviewer_bzhr · 2024-10-28

**Summary Of Contributions:**

The paper proposes Batch-Scheduled Sampling and Reference-Answer-based Correction to reduce the gap between training and inference. The authors conduct sufficient experiments on different datasets to support their claims. Also, the authors have conducted the analysis to visualize the embedding distance between generated and reference responses.

**Audience:**

Yes

**Claims And Evidence:**

Yes

**Requested Changes:**

Please check the weakness.

**Strengths And Weaknesses:**

Strengths:
* The methods are great. It is easy to understand.
* The performance improvement is significant.
* The author validates the method on different datasets, proving its effectiveness.

Weakness:
* Does should $z^j=p_{\theta}(.|x,g^{j})$ be rewritten as  $z^j=argmax~p_{\theta}(.|x,g^{<j})$?
* For Equation 4, could you give an example to explain how you replace $y^j$ with $z^j$?
* In Table 3, BASH performs better than RAC, could you explain the reason?

---

> ### Author Response · Authors · 2024-10-29
>
> Thank you for your feedback. We’re glad that the reviewer recognizes the effectiveness of our method, as demonstrated through extensive experiments.
>
>
> > Should $z^j= p_\theta(\cdot|x,g^j)$ be written as  $z^j=\arg\max p_\theta(\cdot|x,g^{<j})$?
>
> We assume the reviewer is referring to the equation $z^j \sim p_\theta(\cdot|x, g^{<j})$ above Eq. (4). In BASH, we can utilize different sampling strategies. While greedy sampling, as the reviewer suggests, can produce more consistent samples, it lacks diversity and results in generations that are often close to each other, especially when the model is already fine-tuned. The most effective generations occur when samples are sufficiently varied from each other, allowing the model to encounter scenarios that more closely resemble those at inference time. Therefore, instead of selecting the token with the highest probability, we sample from the distribution $p_\theta(\cdot|x, g^{<j})$ to obtain $z^j$. This approach is also adopted in the original scheduled sampling algorithm [1].
>
> [1] Bengio, Samy, et al. "Scheduled sampling for sequence prediction with recurrent neural networks." Advances in neural information processing systems 28 (2015).
>
> > An example of replacing $y$ with $z$.
>
> We provide an example in the following tables. $y$ denotes the ground-truth sentence from the demonstration dataset. $z$ is the model generation based on the previous tokens. $g$ is a mixture of $y$ and $z$. In each step, we first generate $z_i$ by feeding previous tokens from $g$ (**instead of from $z$**) to LM: $z_i \sim p_\theta(\cdot|x,g^{<j})$. For example, to obtain $z_3$, we input query and previous tokens $[g_1,g_2]$ (i.e., "Peter's") to the model and do sampling from the output distribution and get "age".
>
>   | i                  | 1     | 2    | 3       | 4     | 5    | 6    | 7    |
>   | ------------------ | ----- | ---- | ------- | ----- | ---- | ---- | ---- |
>   | $y_i$              | Peter | is   | 30      | years | old  | now  | ,    |
>   | $z_i$              | In    | 's   | **age** |       |      |      |      |
>   | replace $y$ by $z$?| N     | Y    |         |       |      |      |      |
>   | $g_i$              | Peter | 's   |         |       |      |      |      |
>
>   Then we get a mixed sequence $g_3$ by replacing $y_3$ with $z_3$. Suppose the replacement flag (sampled from a Bernoulli distribution) is "Y", then $g_3=z_3$.
>
>   | i                  | 1     | 2    | 3       | 4     | 5    | 6    | 7    |
>   | ------------------ | ----- | ---- | ------- | ----- | ---- | ---- | ---- |
>   | $y_i$              | Peter | is   | 30      | years | old  | now  | ,    |
>   | $z_i$              | In    | 's   | age     |       |      |      |      |
>   | replace $y$ by $z$ | N     | Y    | Y       |       |      |      |      |
>   | $g_i$              | Peter | 's   | **age** |       |      |      |      |
>
>   We repeat the above generation for all steps until the response ends. The final results look like
>
>   | i                  | 1     | 2    | 3    | 4     | 5    | 6    | 7    | ...  |
>   | ------------------ | ----- | ---- | ---- | ----- | ---- | ---- | ---- | ---- |
>   | $y_i$              | Peter | is   | 30   | years | old  | now  | ,    | ...  |
>   | $z_i$              | In    | 's   | age  | is    | old  | and  | .    | ...  |
>   | replace $y$ by $z$ | N     | Y    | Y    | N     | N    | N    | Y    | ...  |
>   | $g_i$              | Peter | 's   | age  | years | old  | now  | .    | ...  |
>
>   As illustrated, the next token prediction of $z$ is conditioned on previous tokens of $g$ instead of $z$. Therefore, the sequence $z$ and $g$ will not significantly deviate from the demonstration $y$ but can still expose the LM to inference distribution gradually. As a result, $z$ and $g$ may not always be smooth due to the nature of their generation.
>
>
>
> > In Table 3, BASH performs better than RAC, could you explain the reason?
>
> That's a great observation. We hypothesize that BASH outperforms RAC here due to the specific characteristics of these benchmarks. For example, Math QA requires mathematical calculations and reasoning, where the gap between training and inference can arise from differing reasoning trajectories. RAC may not always be able to correct the reasoning error, as shown in the failure example provided in Appendix A.4. On the contrary, BASH may address this gap more effectively by considering multiple trajectories during training, which simulates how the model would generate different paths during inference. However, while RAC lags behind BASH in performance on this benchmark, it still improves results compared to other methods. Additionally, the self-correction capability of RAC can be instrumental in tackling more complex scenarios where the search space is large and requires multiple steps of reasoning.

---

### Review · Reviewer_PCM5 · 2024-11-01

**Summary Of Contributions:**

The paper proposes an efficient method to perform scheduled sampling at scale. Prior work has investigated that "teacher-forcing" can result in the LM learning an incorrect distribution of tokens. Scheduled sampling and its variants were proposed as solutions.
The authors claim that scheduled sampling is hard to scale, leading to it not being used while training large LMs.
The paper proposes BASH (Batched Scheduled Sampling), which creates a mixed dataset between teacher-forced and model-generated sequences. The authors also perform BASH between training instances which allows the approach to scale.

The authors note that BASH can result in creating "incorrect" sequences and to account for this, they propose an alternative method called RAC that self-corrects these model-generated sequences post-hoc.
The authors empirically show that BASH and RAC improve model performance on several downstream tasks like summarization, math and general QA, and instruction following.
The authors compare their approach against several baselines notably NEFTtune, SPIN, and vanilla SFT. They find that BASH and RAC outperform.
The authors also analyze the gap between training and inference using the distance of the embeddings of the training and inference sequences and they find that models trained using BASH and RAC exhibit lower distance indicating lesser discrepancy between the two.

**Audience:**

Yes

**Broader Impact Concerns:**

No Broader Impact Statement is present. I don't see any ethical concerns surrounding this work.

**Claims And Evidence:**

Yes

**Requested Changes:**

Overall I feel positive about the paper. I believe that the paper would benefit strongly from a more detailed analysis of what aspects of BASH and RAC improve performance. However, I don't think that this is critical for acceptance.

**Strengths And Weaknesses:**

Strengths

- The paper proposes two simple and effective ideas to scale scheduled sampling.
- The paper is generally well-written and easy to understand.
- The experimentation is extensive -- showing the effectiveness of the approach across several different tasks and models.
- There is some initial analysis into what makes these approaches better.

Weaknesses

- The analysis of why BASH and RAC work seems a bit preliminary and the paper might benefit from a deeper analysis.
- The algorithms are variants of scheduled sampling and the paper does not compare BASH and RAC to older works on scheduled sampling (even on very small models as older scheduled sampling hard to scale)

Questions
- Intuitively it seems that BASH should underperform RAC on the mathematical reasoning datasets as BASH allows the model to be trained on potentially incorrect reasoning chains. But the inverse is observed in the results. Why do you think this is the case?

Formatting errors
- Footnote on page 8 overflows to page 9.

---

> ### Author Response · Authors · 2024-11-13
>
> Thanks for your constructive feedback. We’re glad that the reviewer recognizes the effectiveness of our method.
>
> > The analysis of why BASH and RAC work seems a bit preliminary and the paper might benefit from a deeper analysis.
>
> To demonstrate the effectiveness of our method, we closely follow the setup of previous papers that study SFT-related methods, where they only have access to demonstration data like SPIN [1]. In our paper, we conduct more comprehensive experiments than those papers, considering a wider range of benchmarks, model sizes, and scenarios where preference data (in sec. 4.2.4) is also available. Moreover, we provide a more thorough evaluation compared to previous works by evaluating multiple generations multiple times on the AlpacaEval 2.0 benchmark, which is rarely done in prior studies. Additionally, we report both greedy and sampling-based approaches to show the effectiveness of our method, regardless of how the samples are generated—something that is seldom addressed in previous work. We also have many qualitative analyses and ablations in the appendix.
>
> We are also glad to consider any additional specific experiments that the reviewer thinks will help strengthen our paper.
>
> [1] Self-Play Fine-Tuning Converts Weak Language Models to Strong Language Models, ICML 2024 https://arxiv.org/abs/2401.01335
>
>
>
> > The paper does not compare BASH and RAC to older works on scheduled sampling.
>
> As suggested by the reviewer we run new experiments where we compare the performance of BASH with standard scheduled sampling (SCS), which mixes expert demonstrations with tokens generated by the current model in an online fashion. We test it on a summarization task, keeping all other hyperparameters the same.
>
> |                    | win rate (%) | Rouge 1 | Rouge 2 | Rouge L |
> | ------------------ | ------------ | ------- | ------- | ------- |
> | SFT                | 27.03±0.38   | 31.69   | 11.02   | 24.53   |
> | Scheduled sampling | 28.24±0.46   | 32.03   | 11.22   | 24.95   |
> | BASH               | 28.12±0.43   | 32.00   | 11.27   | 24.89   |
>
> The results show that scheduled sampling (SCS) achieves a similar win-rate to our approach, which is an expected result. However, the key difference is that BASH is computationally efficient and practical for large model training, whereas this is not true for SCS. To provide further perspective on the effectiveness of our approach, we conduct new experiments comparing the computational overhead of SCS and BASH on an 8xA6000 (48G) machine using 1B and 7B models.
>
>
>
> Table 7. Comparison on summarization task with pythia 1B model, global batch size=512
>
> |                              | SCS | BASH (offline data generation) | BASH (training) |
> | ---------------------------- | ------------------ | ------------------------------ | --------------- |
> | Computation time for 1 batch | ~20.8s             | ~0.9s                          | ~4.4s           |
> | Memory usage of each GPU     | ~35GB              | ~17GB                          | ~28GB           |
>
> Table 8. Comparison on general QA task with Mistral-7B model, global batch size=256
>
> |                              | SCS | BASH (offline data generation) | BASH (training) |
> | ---------------------------- | ------------------ | ------------------------------ | --------------- |
> | Computation time for 1 batch | > 3600s            | ~370s                          | ~75s            |
> | Memory usage of each GPU     | ~ 46GB             | ~22GB                          | ~36GB           |
>
>
>
>
> As these results show, SCS requires significantly more GPU memory and computation time, taking much longer time to run/train/compute than BASH.
>
>
>
> > It seems that BASH should underperform RAC on the mathematical reasoning datasets as BASH allows the model to be trained on potentially incorrect reasoning chains. But the inverse is observed in the results.
>
> That's a good observation. Both BASH and RAC can suffer from the issue of training the model on an incorrect reasoning path. RAC may not always be able to correct reasoning error, as shown in the failure example in Appendix A.4. Our hypothesis on why BASH outperforms RAC is that BASH can consider multiple reasoning paths (by generating various tokens $z$ and mixing it with expert demonstration) while RAC mainly trains on self-generated reasoning trajectory and it cannot fully correct complex reasoning error. Though RAC under-performs BASH on math tasks, it still improves the accuracy over other baselines.

---

### Review · Reviewer_upQw · 2024-11-03

**Summary Of Contributions:**

This paper proposed two methods (BASH and RAC) to fine-tune language models (LMs). BASH involves replacing ground-truth tokens with tokens generated by the LM itself, in order to mitigate the discrepancy between training and inference time. To align the self-generated target, RAC was also proposed where the self-generated target is generated with a prompt that includes the reference answer.

**Audience:**

Yes

**Claims And Evidence:**

No

**Requested Changes:**

1. Overhead & performance vs batch size (Weakness 2)
2. Experiments mentioned in weakness 3.

**Strengths And Weaknesses:**

Strengths

The experiment results suggest the proposed methods perform better than a recent  baseline. The proposed batch-scheduled method should be faster than existing methods of the same purpose. Writing is generally clear.

Weaknesses:

1. While I see the motivation behind RAC, I found the token-by-token approach not super convincing. A factual error likely appears very early in the generation - for example, a response to "Did Amy go to school on Monday?" naturally starts with "Yes/No". If the model is forced to learn to complete something that starts with "No" when it *knows* that the reference answer is "Yes" (from the prompt), it looks like also a big distribution shift to me, arguably more than that caused by completing ground-truth vs self-generated prefixes.
	a. Also, while it empirically might have improved the performance, generating withe prompt during training time vs generating without the prompt seem like a bigger discrepancy to me.

2. If I understand correctly, BASH is essentially SCS, but batched. The novelty thus looks low. Also, to show that BASH is actually useful, some experiments on the overhead vs batch-size (i.e. how often do we stop and regenerate the dataset) as well as the corresponding impact on performance is due.

3. SCS was from pre GPT time when LMs were much less performing. Since modern LLMs are pretty strong, the mismatch between training and inference is less obvious, and the motivation of this paper might be weakened. Some analysis/experiment on how the improvement changes as models become bigger should be included.

---

> ### Author Response · Authors · 2024-11-13
> **Response to Reviewer upQw (1/2)**
>
> Thank you for your feedback. It seems that the primary concern raised by the reviewer is that the gap between training and inference is not as important as it was with pre-GPT LMs. We hope that, after seeing our explanations, it will be clearer why this remains a critical problem and how our approaches provide a way to address it. Please let us know if you have any further questions.
>
> > While I see the motivation behind RAC, I found the token-by-token approach not super convincing. A factual error likely appears very early in the generation. For example, a response to "Did Amy go to school on Monday?" naturally starts with "Yes/No". If the model is forced to learn to complete something that starts with "No" when it knows that the reference answer is "Yes" (from the prompt), it looks like also a big distribution shift to me, ...
>
> That is, in fact, the main motivation of our paper, as the auto-regressive model generates token-by-token, which leads to a distribution shift between what the model sees during training time and what it will see during test time. During training, such a shift never occurs because the model always uses the ground truth to guide training. However, during testing, the model uses its own generated token at each time step to generate the next one. Hence, with slightly different generations, the model can end up with a completely different distribution than what was seen before (e.g. reviewer’s example). That is why we studied this problem, and our proposed solutions in this paper address the issue by making training and inference times as similar as possible, thereby making the model more robust to such behaviors. As our results show, our proposed approaches are clearly effective.
>
>
>
>
>
> > Also, while it empirically might have improved the performance, generating with prompt during training time vs generating without the prompt seem like a bigger discrepancy to me.
>
> We are not sure what the reviewer means by this comment. Could you please provide more details so that we can respond appropriately? Our RAC method corrects model’s generation by augmenting with demonstration response. The results clearly demonstrate the effectiveness of the proposed approaches, supporting our claim about their utility.
>
>
>
>
> > ... BASH is essentially SCS, but batched. The novelty thus looks low. some experiments on the overhead vs batch-size (i.e. how often do we stop and regenerate the dataset) as well as the corresponding impact on performance is due.
>
> Yes, that is the entire point of BASH: to make SCS practical for large models. It is an offline variant of SCS, which was not considered in previous works. We would like to note that our goal isn't to invent a new SCS method, but rather to adapt this effective approach (which was effective before large models) for the training of large models. Our results corroborate our claim.
>
> Also, as the reviewer suggested, we conducted new experiments to compare the computational overhead of SCS and BASH on an 8xA6000 (48G) machine using 1B and 7B models. We would like to emphasize that as models get larger, SCS becomes more expensive and ultimately impractical. For larger models, switching between training and inference (to do generation) is highly costly because the model is sharded across multiple GPUs. For instance, in larger models, weights, gradients, and optimizer states are all sharded across several GPUs during training, making it impractical to switch between training and inference for every single token. We refer the reviewer to https://www.deepspeed.ai and https://huggingface.co/docs/transformers/en/performance, which discuss the challenges of distributed training and inference. In our experiments, we perform offline generation only once per epoch (i.e., in each epoch, we first generate BASH sequence for the whole training dataset and then train on that).
>
> Table 7. Comparison on summarization task with pythia 1B model, global batch size=512
>
> |                              | SCS | BASH (offline data generation) | BASH (training) |
> | ---------------------------- | ------------------ | ------------------------------ | --------------- |
> | Computation time for 1 batch | ~20.8s             | ~0.9s                          | ~4.4s           |
> | Memory usage of each GPU     | ~35GB              | ~17GB                          | ~28GB           |
>
> Table 8. Comparison on general QA task with Mistral-7B model, global batch size=256
>
> |                              | SCS | BASH (offline data generation) | BASH (training) |
> | ---------------------------- | ------------------ | ------------------------------ | --------------- |
> | Computation time for 1 batch | > 3600s            | ~370s                          | ~75s            |
> | Memory usage of each GPU     | ~ 46GB             | ~22GB     | ~36GB           |
>
> As these results show, SCS requires significantly more GPU memory and computation time, taking much longer time to run/train/compute than BASH.

---

> > ### Author Response · Authors · 2024-11-13
> > **Response to Reviewer upQw (2/2)**
> >
> > > SCS was from pre GPT time when LMs were much less performing. Since modern LLMs are pretty strong, the mismatch between training and inference is less obvious, and the motivation of this paper might be weakened.
> >
> >
> > We respectfully disagree with the reviewer's comments. Although current LLMs are powerful, the mismatch between training and inference remains an issue. Current LMs are primarily auto-regressive models, and regardless of their size, the way they are trained is different from how they are utilized during inference, hence the gap between training and inference time is still there. While larger size brings advantages, it does not resolve the discrepancy between training and inference by itself alone. The reviewer's own examples and comments clearly highlight the issues with auto-regressive models and their token-by-token generation. Our proposed approaches provide principled ways to address this problem without requiring changes to the architecture or any modifications during inference time. Instead, our approaches make training and inference times as close as possible.
> >
> > We would also like to refer the reviewer to previous work such as [1], where they discuss the problem associated with teacher-forcing in auto-regressive models (which is the source of this discrepancy), yet do not provide a solution. In our work, we not only discuss this problem but also propose practical solutions to address it.
> >
> > [1] The pitfalls of next-token prediction, ICML 2024 https://arxiv.org/abs/2403.06963

---

> ### Comment · Reviewer_upQw · 2024-12-16
>
> Thank you for the response. However, please allow me to clarify my concerns, as it seems like the current response does not address them. My main concern is that the motivation/story is not convincing and cannot support the performance, and while this method might work in general, it seems like it doesn't work for the reasons mentioned in this paper.
>
> Regarding my comments above the token-by-token approach: In my example, if the model's own generation is wrong (i.e. starts with "No" when the reference answer is "Yes"), then it is forced to complete this sentence even though this sentence prefix is a dead end already. From my experience, in most question-answering tasks at least, the first few tokens matter the most. Thus, the token-by-token approach tries to teach the LM to paraphrase a wrong sentence into something that looks like the reference sentence, and I'm not sure why this should help the model's performance at inference time.
>
>
>
>
> > Also, while it empirically might have improved the performance, generating with prompt during training time vs generating without the prompt seem like a bigger discrepancy to me.
>
> By this, I mean that the prompt provided in, for example, footnote 6, is not available at inference time. This seems like a bigger training-inference discrepancy than the one this paper aims to solve.
>
>
> Regarding the mismatch for more modern LLMs, what I meant in this comment is that since their performance is much better, their own generation is less different (in distribution) from the data they are trained on.
> > The reviewer's own examples and comments clearly highlight the issues with auto-regressive models and their token-by-token generation.
>
> My example is strictly related to the proposed method. Please see my comments above.

---

> > ### Author Response · Authors · 2024-12-17
> >
> > Thanks for your clarification. However, we believe there are still some misunderstandings and we respectfully disagree with some of your statements.
> >
> > > If the model's own generation is wrong, ..., the token-by-token approach tries to teach the LM to paraphrase a wrong sentence into something that looks like the reference sentence, and I'm not sure why this should help the model's performance at inference time.
> >
> > The objective of our method is to mitigate the training-inference gap by making the training process more closely resemble inference time and guiding the generation from incorrect paths to correct ones. In these approaches, both correctness and **distribution consistency** are important for the guidance. If we only emphasize the correctness and train model solely on the reference answer, the model will fail to learn how to recover when its generation deviates during inference because we never teach it in that case.
> >
> > In your example, the RAC sequence starting with "No" may be less correct than the reference answer (starting with "Yes") but it is more correct than model's generation. More importantly, it's a good guidance in model training because **it simulates the case of inference** and it has a **consistent** distribution with model inference, whereas the reference answer (which lacks this alignment) does not provide such guidance when the generation deviates.
> >
> > > ..., the prompt provided in footnote 6, is not available at inference time. This seems like a bigger training-inference discrepancy than the one this paper aims to solve.
> >
> > The RAC sequence is used exclusively as a correction mechanism during training and does not rely on the same prompt at inference time. Therefore, it doesn’t introduce any additional training-inference discrepancies. Furthermore, our experiments validate that incorporating RAC sequences during training effectively mitigates the distribution shift between reference answer and model generation both on training and test sets.
> >
> > > Regarding the mismatch for more modern LLMs, what I meant in this comment is that since their performance is much better, their own generation is less different (in distribution) from the data they are trained on.
> >
> > We disagree with it. The very nature of auto-regressive models, in which each token is conditioned on the previous token, can lead to vastly different generations even with slight changes in only a few tokens. The resulting distirbution shift remains a persistent issue in modern LLMs, as illustrated by the related work [1]. In addition, our experiments on the general QA task with a 7B model trained on an extensive dataset (Ultrachat-200k) show that the training-inference gap still exists. Our proposed method effectively reduces this gap, resulting in measurable improvements in performance.
> >
> > [1] The pitfalls of next-token prediction, https://arxiv.org/abs/2403.06963

---

### Author Response · Authors · 2024-11-21
**General response to reviewers**

We thank all of the reviewers for their valuable feedback and we are glad to receive it, and certainly glad to see that reviewers believe our paper to be well-written, well-motivated, simple but effective, the experiments thorough and comprehensive, and its potential.

**Summary of our contributions**:

The common approach to training autoregressive models is teacher forcing. In this method, the ground truth tokens from the previous time steps are used as the input for the model at the current time step. This method can have some drawbacks. Most notably, when a model becomes too reliant on the provided ground truth tokens, it may fail to generalize well to unseen data during inference, where the model uses its own generations. To address this problem, we propose two simple but effective approaches, BASH and RAC, where the model utilizes its own generation during training time in a manner similar to how samples are generated during inference. This provides the model with an explicit self-correction capability. Our extensive experiments and analysis demonstrate the effectiveness of our method across various benchmarks and tasks.

**New experiment results and analysis**:

Following reviewers' suggestions, we have run new experiments and analysis and will include the following in the updated paper:
- Per Reviewer upQw and PCM5 requests, we run new experiments where we compare computational overhead of SCS and BASH (see table 7 and 8 in the responses to the PCM5 and upQw).
-  Per Reviewer PCM5 request, we compare the performance of BASH with SCS (see our response to Reviewer PCM5).

These new results provide further evidence for the effectiveness and applicability of our proposed approaches.

---

### Decision · Action_Editor_2W2c · 2025-01-08

**Recommendation:** Accept with minor revision

**Comment:**

This paper is motivated by the fact that there is a difference between the way models are trained and what they do at inference time; this is a well-known observation. During training, the standard approach is to use the reference/ground-truth tokens for the loss. Naturally these are not available at inference time, so only the generated tokens are used to predict the next token---unlike in training. This gap can lead to potential performance costs. The authors introduce two methods to address the issue. One is a more efficient batch version of the scheduled sampling method. The other is a method to integrate a method for self-correction---a fairly creative approach.

Most reviewers found that there were a number of interesting insights in this paper. The experimental results are particularly solid. One of the reviewers asked for a more detailed analysis of the self-correction approach (or at least more of an explanation for what should happen in certain intuitive cases where it's not clear what the method will do, as spotted by the reviewer). The argument raised is that while the authors' original motivation is specifically to remove the gap between training and inference, in certain cases this need not happen, i.e., the authors' method is producing improved results without strictly removing this gap. While I appreciate the reviewers' thoughtful points, even if this is the case, the fact that the self-correction approach appears to actually deliver strong results makes it worthwhile. In fact this would be an interesting finding on its own.

Ultimately there is enough here that is valuable that I believe the paper is worth accepting. For the camera ready version, the authors are encouraged to provide a bit more commentary on the scenarios brought up above.

**Audience:**

The paper studies training procedures for language models, and is therefore of interest to a large part of the TMLR audience.

**Claims And Evidence:**

Yes, there is sufficient evidence for the claims being made.

---

> ### Author Response · Authors · 2025-01-20
> **Camera-ready version**
>
> We would like to thank the Reviewers and the Action Editor for their valuable suggestions and insightful feedback. We have uploaded the deanonymized camera-ready version of the manuscript. Here are the updates:
> - In appendix A.3 (page 18~19), we add new comparison results between SCS and BASH in terms of final performances and computation overhead, as suggested by Reviewer PCM5.
> - In appendix A.4, we add more discussion on the reference-answer augmented correction (RAC). We provide more explanations on when the RAC will fail and why the RAC fails in those cases, and we also mention the discussions in the main text.